# Microscale Thermophoresis Reveals Oxidized Glutathione as High-Affinity Ligand of Mal d 1

**DOI:** 10.3390/foods10112771

**Published:** 2021-11-11

**Authors:** Soraya Chebib, Wilfried Schwab

**Affiliations:** Biotechnology of Natural Products, Technical University Munich, Liesel-Beckmann-Str. 1, 85354 Freising, Germany; soraya.chebib@tum.de

**Keywords:** Mal d 1, ligand, glutathione

## Abstract

Pathogenesis-related (PR)-10 proteins, due to their particular secondary structure, can bind various ligands which could be important for their biological function. Accordingly, the PR-10 protein Mal d 1, the major apple allergen, probably also binds molecules in the hydrophobic cavity of its secondary structure, but it has not yet been investigated in this respect. In this study, various natural products found in apples such as flavonoids, glutathione (GSH), and glutathione disulfide (GSSG) were investigated as possible ligands of Mal d 1 using microscale thermophoresis. Dissociation constants of 16.39 µM, 29.51 µM, 35.79 µM, and 0.157 µM were determined for catechin, quercetin-3-*O*-rhamnoside, GSH, and GSSG, respectively. Molecular docking was performed to better understand the underlying binding mechanism and revealed hydrophobic interactions that stabilize the ligands within the pocket while hydrophilic interactions determine the binding of both GSH derivatives. The binding of these ligands could be important for the allergenicity of the PR-10 protein and provide further insights into its physiological role.

## 1. Introduction

In North America, and Central and Northern Europe, most immediate allergic reactions are triggered by pollen and food allergens of the Bet v 1 family, named after the major pollen allergen of white birch (*Betula verrucosa*), which causes seasonal allergies. The structural Bet v 1 homologue Mal d 1 is one of the major allergens in apple (*Malus domestica* L. Borkh), and causes immunological cross-reactions in Bet v 1 sensitized patients [1]. Sensitization to Mal d 1 is associated with symptoms ranging from mild and local allergic reactions (oral allergy syndrome) to severe reactions such as anaphylactic shock [2]. Since Mal d 1 is a heat labile and protease sensitive protein, the allergic potential can be altered by various processing procedures such as heating [3,4,5]. Mal d 1 belongs to the family of pathogenesis-related (PR)-10 proteins and their expression is induced by various environmental conditions including pathogens (e.g., bacteria, fungi), wounding, UV light, and chemicals [6]. However, PR-10 proteins are also constitutively expressed in certain plant development stages [7] and they occur in a variety of different isoforms, caused by gene duplication or by recombinant coding of multiple genes [8]. The characteristic PR-10 fold consists of a seven-stranded antiparallel β-sheet, two short α-helices in V-shapes, and a long C-terminal α-helix [9]. The arrangement of these secondary structural elements creates a large, solvent accessible hydrophobic cavity capable of binding hydrophobic ligands [10]. The affinity properties between a protein and different interaction partners can have a significant impact on its biological activity. However, the biological function of Mal d 1 and other PR-10 proteins is still poorly understood [10]. The sequence homology of PR-10 proteins with ribonucleases from ginseng callus suggested ribonuclease activity for the entire protein family that could be affected by phosphorylation [11,12]. Multiple studies revealed the binding of several hydrophobic ligands such as cytokinins, phytosteroids, and flavonoid glycosides to members of the Bet v 1 family, suggesting binding, transport, and storage functions of the protein family (Table 1) [13,14,15,16,17]. Phytosteroid binding was reported for Pru av1, the homologue protein in cherry [18], but was also suggested for Bet v 1 as both proteins show high structural similarity to the START domain of the human protein MLN64, that is considered to be a cholesterol transfer protein [19]. The crystal structure of Bet v 1 isoforms a and l could be elucidated as complexes with 1-anilino-8-naphthalene sulfonate (ANS) and with the buffer steroid deoxycholate, respectively [20,21]. In addition, Bet v 1 has the ability to bind and permeabilize membranes, suggesting a mechanism for sensitization by and the allergic response of Bet v 1 [22].

The aim of this study was to explore for the first time the interaction of recombinant rMal d 1.02 and various potential ligands using microscale thermophoresis (MST). The MST instrument detects a change in the fluorescence signal along a temperature gradient induced by an infrared laser [28]. The thermophoresis of the fluorescent target protein is different from that of the protein–ligand complex as the binding event results in changes of the hydration shell, charge, and/or size. An equilibrium dissociation constant (K_d_) can be determined by using a dilution series of, usually, 16 MST measurements with a constant amount of fluorescent binding partner and increasing amounts of nonfluorescent ligand. We were able to identify for the first time a high affinity interaction between our target protein rMal d 1.02 and oxidized glutathione (GSSG). We further detected the binding of reduced glutathione (GSH; γ-Glu-Cys-Gly), (+)-catechin, (+/−)-epicatechin, and quercetin-3-*O*-rhamnoside. Molecular docking was used to investigate the underlying interaction mechanism.

## 2. Materials and Methods

### 2.1. Chemicals

The chemicals were purchased from Carl Roth (Karlsruhe, Germany), Sigma-Aldrich (Taufkirchen, Germany), and Merck (Darmstadt, Germany), unless otherwise stated in the text. The reference substances quercetin-3-*O*-rhamnoside and (+)-epicatechin were obtained from our in-house standard reference library.

### 2.2. Production and Purification of Recombinant rMal d 1

Recombinant rMal d 1 was produced in *E. coli* BL21 (DE3) pLysS, as previously described [29]. A single colony was inoculated into 25 mL of LB medium, containing 100 µg/mL ampicillin and 34 µg/mL chloramphenicol. Cells were cultured overnight at 37 °C under constant shaking. Fifteen mL overnight culture was inoculated into fresh LB medium (1 L) with respective antibiotics (100 µg/mL ampicillin, 34 µg/mL chloramphenicol). Cells were grown at 37 °C to an optical density (OD) of 0.6, and protein expression was induced with isopropyl-β-D-1-thiogalactopyranoside (IPTG; 100 µg/mL) at 18 °C overnight. The suspension was centrifuged (15 min at 4 °C and 5000× *g*). Pellets were frozen at −80 °C for 30 min and resuspended in binding buffer (50 mM Tris, 220 mM NaCl, 10 mM imidazole, pH 7.5) containing 0.5 mM phenylmethylsulfonyl fluoride (PMSF), 1 mM MgCl_2_ (New England Biolabs, Frankfurt, Germany), 1 µL DNAse and 0.5 mg/mL lysozyme, followed by cell disruption using ultrasonication (3 × 1 min, cycle 5 × 10%, 50% power). The soluble protein fraction was applied onto a column containing His-Tag resin (Bio-Rad Laboratories, Feldkirchen, Germany) and incubated overnight at 4 °C. The soluble protein fraction was purified by Profinity Immobilized Metal Ion Affinity Chromatography (IMAC) using 10 mM to 250 mM imidazole. The purified fractions were dialyzed against sodium carbonate buffer (10 mM Na_2_CO_3_, 10 mM NaHCO_3_, pH 9), lyophilized, and stored at −20 °C until further use. Protein was resuspended in PBS buffer (150 mM NaCl, 10 mM Na_2_HPO_4_, pH 7.4 adjusted by 10 mM NaH_2_PO_4_) and incubated at 4 °C for at least 2 h prior to analysis. Protein concentrations were measured by UV/Vis spectrometry using an extinction coefficient of 0.849 [30,31].

### 2.3. Microscale Thermophoresis

The protein rMal d 1 was labeled using the His-Tag labeling kit RED-tris-NTA 2nd Generation (Nanotemper Technologies GmbH, Munich, Germany). The labeling was performed according to the protocol provided by the manufacturer. Proteins were diluted to the final concentration of approximately 960 nM in PBS-T buffer (150 mM NaCl, 10 mM Na_2_HPO_4_, pH 7.4, adjusted by 10 mM NaH_2_PO_4_, 0.05% Tween 20). RED-tris-NTA was diluted in PBS-T to a final concentration of 100 nM. Then, 90 µL of protein (~960 nM) was mixed with 90 µL of dye (100 nM). The mixture was incubated on ice for 30 min in the dark. The sample was centrifuged for 10 min at 4 °C and 15,000× *g* and the supernatant transferred to a fresh tube. MST measurements were performed with rMal d 1 and the following substances: reduced glutathione (GSH), oxidized glutathione (GSSG), glycine, glutamic acid, quercetin-3-*O*-rhamnoside, (+)-catechin, and (+/−)-epicatechin. GSH, GSSG, glycine, and glutamic acid were dissolved in PBS buffer, while quercetin-3-*O*-rhamnoside was dissolved in carbonate/bicarbonate buffer (30 mM Na_2_CO_3_, 70 mM NaHCO_3_, pH 9.6). (+)-Catechin and (+/−)-epicatechin were dissolved in DMSO. All ligands were further diluted in PBS-T; organic solvent concentrations in each assay were kept below 5 %. A total of 16x 1:1 ligand dilutions in an end volume of 10 µL were prepared in PBS-T buffer. The final concentration ranged for GSSG from 100 µM to 3.05 nM and from 500 µM to 15.25 nM; for GSH, (+)-catechin and (+/−)-epicatechin from 0.001 µM to 30.52 nM; for quercetin-3-*O*-rhamnoside from 500 µM to 15.25 nM. The amino acids glutamic acid and glycine were used in a concentration range from 12 mM to 366 nM and 24.29 mM to 741.27 nM, respectively. Ten µL of labeled rMal d 1 was added to all dilutions. Reactions were incubated on ice for at least 30 min and then loaded into Monolith NT.115 capillaries. All measurements were carried out with a Monolith NT.115 device (Nanotemper Technologies GmbH, Munich, Germany), using the MO.Control software v1.6.1, at 60 % LED and high MST power. In each analysis, it was ensured that the intensity of the fluorescent target molecule was above >200 counts with variations <20 % and that no protein aggregation occurred. Data was analyzed using MO.Affinity analysis software v2.3 (Nanotemper Technologies GmbH, Munich, Germany). Single outliers were removed from the performed technical replicates if it greatly lowered the standard deviation of the determined K_d_. For GSSG, two technical replicates for each concentration range were independently pipetted and the measurements were analyzed using the signal from MST-on time of 20 s. For GSH and quercetin-3-*O*-rhamnoside, four and three technical replicates, respectively, were measured and analyzed using the signal from MST-on time of 10s. Three replicates of catechin were measured and analyzed using the signal from MST-on time of 20 s. (+)-Epicatechin and (−)-epicatechin, each analyzed in triplicates, were evaluated at an MST-on time of 20 s and 2.5 s, respectively. Glutamic acid and glycine were analyzed by a single run at an MST-on time of 20 s. The MST-on time, which yields a signal-to-noise ratio > 6 in the binding curve, was used to determine the dissociation constant (K_d_). According to the manufacturer’s protocol, GSH interferes with the labeling dye at concentrations above 10 mM. Control experiments with a labeled control peptide (His6 peptide; provided by the manufacturer) and GSH, GSSG, and cysteine were performed to analyze possible interactions with the labeling dye. The lyophilized control peptide was suspended and further diluted in PBST-T to a final concentration of 200 nM. RED-tris-NTA was diluted in PBS-T to a final concentration of 100 nM. Then, 90 µL of the peptide (200 nM) was added to 90 µL of the dye (100 nM) and incubated for 30 min in the dark. A 16x 1:1 dilution series was prepared as described above. The final concentration for GSH and cysteine ranged from 0.001 µM to 30.52 nM and for GSSG from 0.002 µM to 61.04 nM. GSH, cysteine, and GSSG were analyzed in a single run at an MST-on time of 5 s, 5 s and 1.5 s, respectively. The K_d_ was calculated by Equation (1) according to the law of mass action, where f (conc) is the fraction bound at a given ligand concentration; Unbound is the response value of the unbound state; Bound is the response value of the bound state; TargetConc is the final concentration of the labeled molecule (https://www.manualslib.com/manual/1556718/Nano-Temper-Monolith-Nt-115.html; accessed on 10 November 2021).
(1)$$f\left(\mathrm{conc}\right)=\;\mathrm{Unbound}+\frac{\left(\mathrm{Bound}-\mathrm{Unbound}\right)\times (\mathrm{Conc}+\mathrm{TargetConc}+K_{d}-\sqrt{{\left(\mathrm{Conc}+\mathrm{TargetConc}+K_{d}\right)}^{2}-4\times \;\mathrm{Conc}\;\times \;\mathrm{TargetConc}}}{2\times \;\mathrm{TargetConc}}\;$$

### 2.4. Molecular Docking

For molecular docking, AutoDock Vina v1.1.2 (http://vina.scripps.edu/; accessed on 22 September 2021) [32] was used to dock the ligands GSSG, GSH, (+)-catechin, (+/×)-epicatechin, and quercetin-3-*O*-rhamnoside into the hydrophobic pocket of rMal d 1.02. The recombinant protein was modeled using SWISS-MODEL (https://swissmodel.expasy.org/interactive; accessed on 26 July 2021). Input files of rMal d 1 and ligand were created with AutoDockTools v1.5.6 (http://autodock.scripps.edu/resources/adt; accessed on 22 September 2021). The protein structure was prepared for molecular docking by removing all water molecules, computing Kollman charges, and adding polar hydrogens. The hydrophobic pocket of the protein was obtained by BiteNet (https://sites.skoltech.ru/imolecule/tools/bitenet; accessed on 4 August 2021) [33]. The grid box (36/36/34) was placed over the hydrophobic pocket of the protein. AutoDockVina was run by default settings and generated nine possible conformations of the ligand in complex with rMal d 1. Only the ligand conformation with the highest calculated affinity to the protein is presented. Binding energies (∆G) were used to calculate an equilibrium K_d_ by Equation (2) with R = 1.986 cal/mol*K and T = 298.15 K [14]. The resulting models were visualized by Discovery Studio v21.1.0.20298 (https://discover.3ds.com/discovery-studio-visualizer-download; accessed on 22 September 2021).
(2)$$K_{d}={\;e}^{-\frac{\Delta G}{R\times T}}\;$$

## 3. Results

### 3.1. rMal d 1.02 in Complex with Flavonoids

Although the ligand-binding ability of several PR-10 proteins has been recently analyzed (Table 1), Mal d 1 has not been evaluated for this property until now. Since PR-10 proteins are known to bind flavonoids, we selected similar metabolites that occur naturally in apples, namely quercetin-3-*O*-rhamnoside, (+)-catechin, and (+) and (−)-epicatechin as potential candidates for MST binding assays [34]. All binding assays were exclusively carried out using the recombinant rMal d 1. The results of the MST experiments show that these natural flavonoids bind to rMal d 1 with different affinities (Figure 1a,b; Appendix A). (+)-Catechin and quercetin-3-*O*-rhamnoside exhibit affinities in the low micromolar range with K_d_ values of 16.39 ± 5.35 µM and 29.51 ± 7.75 µM, respectively (Figure 1a,b). The enantiomers (+)-epicatechin and (−)-epicatechin bind in the higher µM range with K_d_ values of 152.53 ± 31.03 µM and 663.88 ± 393.46 µM, respectively (Appendix A). Both binding curves do not reach saturation in the bound state, and thus the K_d_ values show higher standard deviations. The S/N of the experiments was above 10, indicating excellent assay conditions and good data quality. In general, the MST experiments showed some variations between the individual technical replicates. This could be due to the occurrence of additional specific or nonspecific binding events that may have directly affected the binding affinity of a compound to the protein. Previous crystallographic studies of other PR-10 proteins reported the binding of additional ligands outside of the pocket, at intermolecular sites [35].

### 3.2. rMal d 1 Binds Glutathione in Both Reduced and Oxidized Forms with Different Specificities

In addition to flavonoids, we studied the binding behavior of GSH and GSSG to Mal d 1, as apple is a rich source of GSH [36]. MST experiments revealed specific interactions between rMal d 1 and GSH, as well as rMal d 1 and GSSG. GSH binds to rMal d 1 with a K_d_ of 35.79 ± 12.76 µM (Figure 1c). The binding of GSSG to rMal d 1 was evaluated by two concentration ranges, of which the dose response curve with 500 nM GSSG as the highest concentration led to a K_d_ of 0.480 ± 0.163 µM (Figure 1d), whereas the binding curve with 100 nM GSSG as the highest concentration provided a K_d_ of 0.157 ± 0.087 µM (Figure 1e). The latter showed an even equilibrium between the bound and unbound state, and thus resulting in a more accurate K_d_ with lower standard deviation. Furthermore, the affinity of the individual amino acids (glycine, glutamic acid, and cysteine), present in GSH, to rMal d 1 was investigated to clarify whether there is a specificity for the entire molecule or only for certain amino acids. The binding of cysteine could not be investigated, as the ligand interfered with the His labeling dye (Appendix A). Interferences between the dye and GSH and GSSG were not observed (Appendix A). No binding was detected for glutamic acid, and only a weak interaction was observed for glycine and rMal d 1 (Appendix A), indicating that the binding of GSSG and GSH to rMal d 1 is specific.

### 3.3. Molecular Docking of the Identified Protein-Ligand Complexes

In addition to the biochemical binding studies, we performed molecular docking of the identified Mal d 1 ligands to the homology model of rMal d 1.02. The structure of Mal d 1 consists of a large internal cavity containing mainly hydrophobic residues, together with some polar and charged side chains forming an amphiphilic binding pocket [37]. The cavity itself can be reached by two openings, one at the N-terminal half of the long helix α3 and one at the side of the β-sheet between the long C-terminal helix α3 and strand β1 (Figure 2 and Appendix A).

The crystal structures of PR-10 proteins showed that the interior of the cavity is filled with ligands, solvent molecules, and buffer components [35]. The presence of water molecules in the hydrophobic cavity plays an important role in protein–ligand complexes by, for example, mediating hydrogen bonding interactions through the presence of solvent accessible polar sites. The binding of flavonoids to PR-10 proteins indicates only one valid binding site within the hydrophobic pocket [13,20,25]. For Fra a proteins from strawberry, it was shown that the loop regions surrounding the hydrophobic cavity are flexible and ligand binding causes conformational changes in the loop regions [13].

The structures of the rMal d 1-flavan/flavonoid complexes obtained via molecular docking (Figure 2a,b) provide possible properties of the binding mechanism. The molecules inside of the Mal d 1 cavity are stabilized by hydrophobic interactions, namely π-sigma and π-alkyl bonds, as well as polar interactions such as hydrogen bonds and van der Waals forces, creating a strong cohesive environment that stabilizes the resulting complex formation. The involved residues in hydrogen bonding of the catechin (Figure 2a) molecule are Ala92 (located in the loop connecting the β5 and β6-strands; Appendix A), Val134 (α3-helix), Tyr102 (β6-sheet), and Gly137 (α3-helix). The (+)-epicatechin interaction is formed with the residues Phe24 (located in the loop connecting the α1 and α2-helices), Lys138 (α3-helix), and Ala141 (α3-helix) in the binding site of the receptor (Appendix A). Six interactions with Ala92 (located in the loop connecting the β5 and β6-strands), Val100 (β6-sheet), His133 (α3-helix), Val134 (α3-helix), Gly137 (α3-helix), and Ala141 (α3-helix) were identified within the (−)-epicatechin–rMal d 1 complex (Appendix A).

The quercetin-3-*O*–rhamnoside complex is stabilized by hydrogen bonding created by the amino acid residues His133 (α3-helix) and Lys138 (α3-helix). The stability of the flavonoid complexes is also attributed to π-sigma and π-alkyl interactions between the aromatic ring of the flavonoid compound and the respective amino acid residues (Figure 2b). These hydrophobic interactions may facilitate the transfer of the molecule into the hydrophobic cavity of the protein.

The high binding affinity between rMal d 1 and GSSG can be linked in particular to the presence of seven hydrogen bonds with the respective amino acids: Phe24 (located in the loop connecting the α1- and α2-helices), Tyr83 (β5-sheet), Lys86 (β5-sheet), Lys138 (α3-helix), and Ala141 (α3-helix) (Figure 3a). The latter also shows an alkyl interaction with the disulfide bond of the molecule, which helps intercalating the ligand, and thus might be responsible for the conformation of the GSSG inside of the protein cavity. The residue Tyr83 was already shown to be a key residue involved in hydrogen bonding in the Bet v 1a–naringenin complex [14,25] and in the Bet v 1–ANS complex [20]. In contrast, the GSH complex is only stabilized by three hydrogen bonds with the respective residues, Tyr85 (β5) and Ala141 (α3) (Figure 3b), thus explaining the lower affinity of GSH to rMal d 1. Both complexes are further stabilized by van der Waals forces with the surrounding residues located in the interior of the cavity.

Molecular docking revealed theoretical K_d_ values of 0.81 µM, 0.35 µM, 0.25 µM, 0.69 µM, 1.35 µM, and 14.35 µM for catechin, quercetin-3-*O*-rhamnoside, (+)-epicatechin, (−)-epicatechin, GSSG, and GSH, respectively (Figure 2 and Figure 3, Appendix A). The K_d_ values for all ligands studied by molecular docking differed to varying degrees from the experimental data. Therefore, it should be noted that due to the large size of the protein cavity, a variety of different orientations of the docked compounds is possible, which could lead to the discrepancy between the experimental and calculated values. Although the docking experiments provided insights into the molecular binding mechanism, further studies are needed to investigate the physiological role of the detected binding interaction and to identify the exact binding region.

## 4. Discussion

### 4.1. Binding of Natural Flavonoids

The flavonoids analyzed in this study as binding partners of Mal d 1 have been isolated from apple flesh and peel [38]. Flavonols and their glycosides are synthesized at the cytosolic side of the endoplasmic reticulum and subsequently accumulated in the central vacuole [39]. The synthesis of polyphenols and Mal d 1 is strongly affected by several stress conditions [40]. Only recently, a significant positive correlation between the content of Mal d 1 and flavan-3-ols has been reported [29]. Polyphenols exhibit antioxidative properties, thus preventing oxidative decomposition of other compounds and ensuring an oxidative equilibrium of reactive oxygen species (ROS) [41]. The binding of these compounds may not only affect their bioavailability, but could also lead to the stabilization of the protein or affect its function [13]. Previous studies have shown that the different PR-10 proteins can bind different ligands (Table 1). The hazel allergen Cor a 1 is associated with the ligand quercetin-3-*O*-(2-*O*-β-D-glucopyranosyl)-β-D-galactopyranoside [26]. In strawberry, the binding of quercetin-3-*O*-glucuronide, myricetin, and (+)-catechin to Fra a1E, Fra a 2, and Fra a 3 was reported with affinities of 5.3 µM, 19.5 µM, and 8.9 µM, respectively, which demonstrated that even isoallergens can vary in their ligand binding specificities (Table 1) [13]. For the PR-10 protein Fra a 1a, transport function for intermediate compounds or products of the flavonoid biosynthesis was postulated [42]. Fra a 1 might also act as signaling component that regulates the flavonoid biosynthesis and metabolite transport through the binding of respective polyphenols [13,42]. Therefore, it is possible that Mal d 1 proteins act as transporter units that regulate the biosynthesis of polyphenols and are thus involved in defense mechanisms. In birch pollen, an isoform specific binding behavior of flavonoids was observed for Bet v 1, the major birch pollen allergen [14,25]. The UV/Vis analysis revealed the binding of quercetin to Bet v 1a, m, and d, with affinities of 9.2, 26.5, and 10.2 µM, respectively (Table 1). The NMR analysis further proved the binding of quercetin-3-*O*-galactoside to Bet v 1 a and m with K_d_ values of less than 5 µM, whereas no binding was observed to Bet v 1 d. A high affinity complex was identified for Bet v 1a and quercetin-3-*O*-sophoroside with a K_d_ of 0.57 µM as determined by fluorescence, suggesting a protective function against UV damage [14]. The authors also demonstrated that the sugar moiety significantly affected the affinity and specificity of the binding, and stereochemical changes of the sugar moiety in the flavonoid compound resulted in different binding affinities to Bet v 1 isoforms. Moreover, the stereochemistry of the aglycon itself affected the binding affinity to the protein. We determined significantly different K_d_ values for the diastereomers (+)-catechin/(+/−)-epicatechin (Figure 1a,b and Appendix A) and even for the enantiomers (+)-epicatechin/(−)-epicatechin (Appendix A). To date, it is unknown whether and how ligand binding and the resulting conformational change affects the allergenicity of an allergen. It is evident that although PR-10 proteins have an identical secondary structure, they still show differences in their binding behavior of physiological relevant ligands. Further studies are needed to investigate the ligand binding behavior of various low and high IgE binding Mal d 1 isoforms, as they can differ in their surface based amino acid residues of the hydrophobic cavity.

### 4.2. Relationships of PR-10 Proteins, Glutathione, and Glutathione S-Transferase Expression

GSH is a tripeptide composed of glutamic acid, cysteine, and glycine, and can be isolated from various apple tissues, with high levels found in apple peel [36]. The thiol can undergo several redox reactions and is converted to its oxidized form, GSSG. GSSG can be reduced back by glutathione reductase using NADPH [43]. The ratio of GSH/GSSG in apple peel is reported to be up to seven, confirming that the greater part of GSH is present in the reduced state [36]. The antioxidant GSH not only takes part in storage and transport processes, but also plays a key role in the detoxification of xenobiotics and reactive oxygen species (ROS) [44]. The latter ones are induced as result of abiotic stress factors and cause severe oxidative damage, such as the inactivation of enzymes. High temperature and excessive solar radiation induce high levels of GSH in apples [45]. On the other hand, increased levels of glutathione can induce the production of plant hormones such as abscisic acid, auxin, and jasmonic acid [46]. GSH can lead to post-translational modifications of reactive S-cysteine residues within proteins for protection against oxidation by S-glutathionylation, as was shown for the PR-10 c protein in birch [47]. Further translational modifications of Bet v 1 isoforms have not been reported. The S-glutathiolation of rMal d 1 by GSH can be excluded because rMal d 1.02 does not contain cysteine residues (Appendix A). Some researchers have postulated that ABC transporters, such as the maize protein MRP1, are involved in the accumulation of flavonoids through the possible formation of GSH conjugates [48]. Previously it was reported that the transport of the anthocyanin, malvidin-3-glucoside into yeast vacuoles overexpressing the ABC grapevine protein, ABCC1, is dependent on free GSH [49]. A recent study confirmed that the uptake of cyanidin-3-glucoside by an ABC transporter is GSH-dependent, but the formation of a GSH conjugate was not shown [50]. In plants, GSH conjugates with anthocyanins or polyphenols in general have not been reported so far [48]. It is possible that GSH and GSSG act as important co-factors by binding to rMal d 1 and navigating the transport of flavonoids to the target cell compartments of the plant, making them available to biosynthetic plant enzymes. Moreover, previous studies reported a high impact of GSH on protein synthesis, such as the transcription of chalcone synthase [51]. In plants, the formation of anthocyanins/ flavonoids, the expression of PR-10 genes, and the synthesis of PR-10 proteins are initiated in response to biotic and abiotic stress [12,35]. Since PR-10 proteins can bind flavonoids, it was suggested that they are also involved in the regulation of the flavonoid biosynthesis.

In a preliminary study, it was found that in white strawberries that were devoid of anthocyanins, the strawberry allergen Fra a 1 was downregulated, along with the enzymes chalcone synthase, dihydroflavonol reductase, and flavonone-3-hydroxylase, which are involved in the flavonoid pathway, compared to red strawberries (Figure 4) [52]. In addition, the RNAi-mediated downregulation of Fra a gene expression in transiently transformed strawberries led to decreased expression levels of flavonoid genes, and caused the formation of white transformed strawberry fruits, thus proposing that *Fra a* genes are of significant importance for the color formation (Figure 4) [42]. Besides, previous studies have shown that the flavonoid transport to the vacuole is driven by the flavonoid binding to glutathione-S-transferase (GST), which is considered to be a cytoplasmic flavonoid carrier protein in vivo (Figure 4) [48,53,54,55]. GSTs are enzymes that catalyze the conjugation of GSH to electrophilic molecules and play an important role in the transport of hydrophobic compounds (Figure 4) [56,57]. Transcriptomic analysis revealed the importance of the expression of the anthocyanidin glucosyltransferase gene and a glutathione S-transferase for the formation of anthocyanins in strawberries (Figure 4) [58]. In maize, the loss of GST function resulted in a total decrease of anthocyanins and further caused the loss of color [59]. Similarly, previous studies have further proposed that the loss of GST function can cause a reduction in anthocyanidin accumulation in concert with the loss of color development in other plants, such as MdGST in apple, VviGST4 in grape, LcGST4 in litchi, and RAP in strawberry [60,61,62,63]. More recently it was shown that the stable downregulation of the PR-10 protein Fra a 1 triggered decreased GSH concentration and lower *GST* expression levels in the transgenic fruits (Figure 4) [64]. As additional endorsement, our results demonstrate the binding of GSH and GSSG to the PR-10 protein Mal d 1, which may be of significance for the regulation of the anthocyanin/flavonoid accumulation in apples.

### 4.3. Effect of Ligand Binding on the Allergenicity of the Protein

Mal d 1 consists of various isoforms with different allergenic properties and distinct ligand binding behavior [65]. It needs to be noted that the ligands identified for recombinant rMal d 1.02 in this study are most probably not entirely the same as those of naturally occurring Mal d 1.02 and other isoforms, as different ligand binding behaviors have already been observed for Bet v 1 isoforms [14,25]. The protein can respond in different ways to the binding of various ligands. Proteins can be stabilized and thus retain their native fold, or can be destabilized and unfold, and they are also able to form aggregates which can be associated with an altered allergenic potential [66,67]. Therefore, the binding of the identified ligands may be able to suppress or increase immunogenicity of rMal d 1. The existence of hypoallergenic and hyperallergenic Mal d 1 isoforms is based on the differences in surface-based amino acid residues that are important for the formation of IgE binding epitopes [68,69]. The glycine-rich region with the key residue Glu-45 was identified as part of a conformational epitope involved in IgE binding in Bet v 1 homologous proteins [70]. Other amino acid residues, such as Thr-10, Thr-57, Ser111, and Thr-112 were also found to play an important role in IgE binding of Mal d 1 [71,72]. In a preliminary study, the binding of zeatin to the PR-10 peach allergen Pru p 1 did not result in any conformational changes of the glycine-rich region [73]. The binding of deoxycholate was reported to stabilize Bet v 1 without modulating its conformational epitopes, suggesting that patients are normally exposed to both ligand-bound and unbound Bet v 1 during the sensitization phase [74]. For Bet v 1 and Fra a 1, the ligand binding led to compact protein structures with less structural flexibility [13,75]. Dynamic protein structures may be important for an optimal configuration of epitope residues and the FcϵRI cross-linkage [75,76]. Recently, a study of the hazelnut allergen Cor a 1.04 isoforms reported an inverse relation between the conformational flexibility and IgE binding [77]. The IgE reactivity correlated with the rigidification of the protein backbone, where the highest IgE binding was observed for Cor a 1.0401, with the most rigid backbone scaffold, whereas the lowest IgE binding potential was determined for the most structurally flexible isoform Cor a 1.0404. Different structural flexibilities have also been shown for Bet v 1 isoforms, of which Bet v 1.0102 has a dynamic protein structure and a low IgE binding capacity in contrast to the more rigid Bet v 1.0101 with a high IgE reactivity [78,79,80]. However, it is widely unknown whether and how the ligand binding to Bet v 1, Mal d 1, and other PR-10 proteins effects the allergenicity. Further studies into the relationship between ligand binding and secondary structure changes and effects on the allergenicity of Mal d 1 are required.

## 5. Conclusions

In this study, we investigated the interaction of several natural flavonoids, as well as GSH and GSSG with recombinant rMal d 1.02, by MST in combination with molecular docking to find possible binding sites. The interactions with the K_d_ values in the high nanomolar range were determined for GSSG, and in the low µM range for catechin, quercetin-3-*O*-rhamnoside, and GSH. While flavonoids are bound via hydrophobic and polar interactions, the binding of GSSG and GSH occurs exclusively via hydrophilic hydrogen bonds and van der Waal forces. Thus, for the first time, the binding of glutathione to a PR-10 protein was demonstrated, further substantiating the importance of this protein family for glutathione S-transferase-mediated anthocyanin accumulation in fruit. The obtained results lay the basis to analyze in detail in follow-up studies the natural ligands of Mal d 1 proteins in different plant parts, and to investigate the significance of the promiscuity of Mal d 1 ligand binding.

## Figures and Tables

**Figure 1 foods-10-02771-f001:**
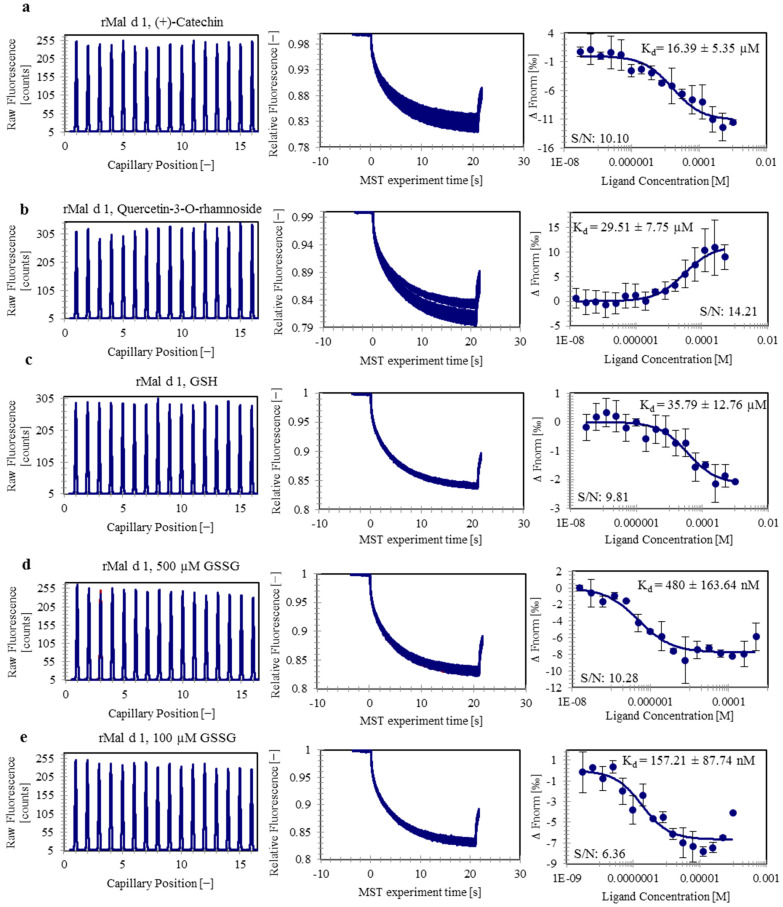
Capillary scan (left), MST traces (middle), and dose–response curves (right) of labeled rMal d 1 bound with catechin (**a**), quercetin-3-*O*-rhamnoside (**b**), GSH (**c**), GSSG (I) (**d**), and GSSG (II) (**e**). Error bars indicate the standard deviation between the performed technical replicates. Dissociation constants (K_d_) and signal-to-noise ratios (S/N) are shown.

**Figure 2 foods-10-02771-f002:**
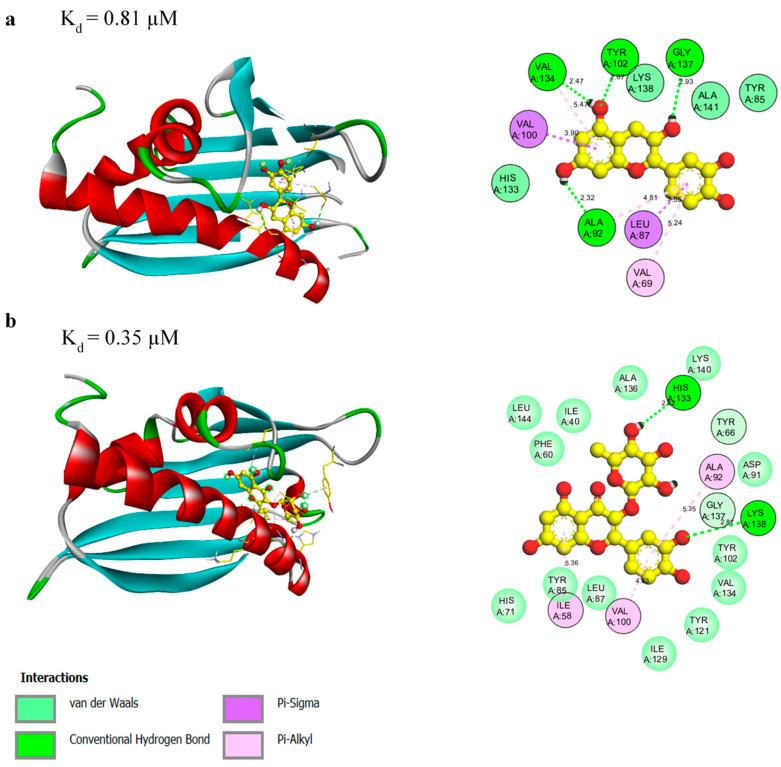
3D (**left**) and 2D (**right**) interaction models of rMal d 1 in complex with (+)-catechin (**a**) and quercetin-3-*O*-rhamnoside (**b**).

**Figure 3 foods-10-02771-f003:**
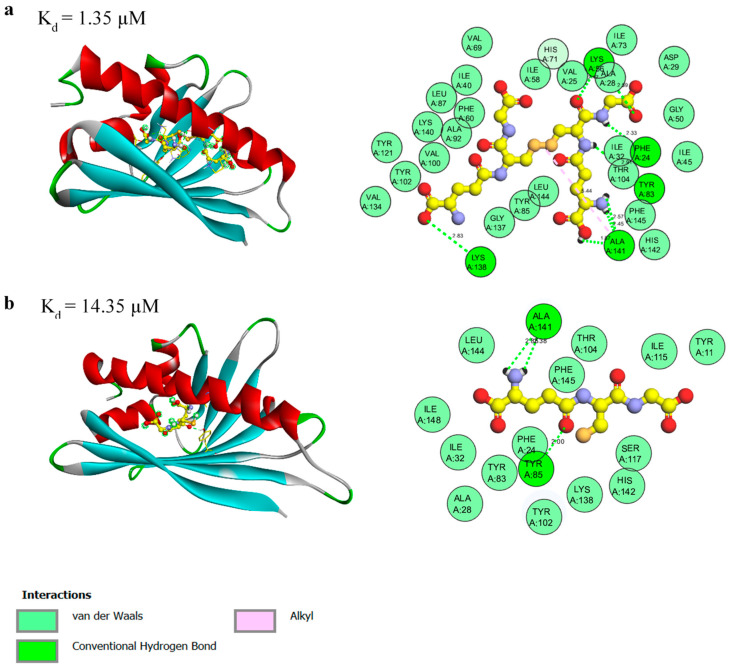
3D (**left**) and 2D (**right**) interaction model of rMal d 1 in complex with GSSG (**a**) and GSH (**b**).

**Figure 4 foods-10-02771-f004:**
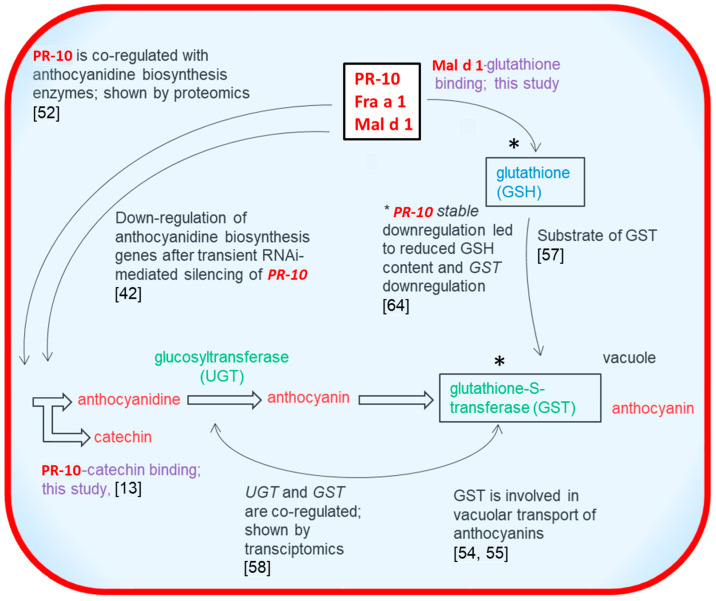
Relationships of PR-10 proteins, flavonoid/anthocyanin biosynthesis genes/enzymes, glutathione content, and glutathione S-transferase expression as reported and shown in this study. * *PR*-*10* stable downregulation led to reduced GSH content and *GST* downregulation [64].

**Table 1 foods-10-02771-t001:** Identified ligands of PR-10 proteins.

Plant	Protein	Ligand	Kd [µM]	Method	References
Birch	Bet v 1a	ANS	18.5	NMR, Fluorescence	[20,23]
		Deoxycholate	58.8	NMR, SAW	[20,24]
		Quercetin	31.4	NMR	[14,25]
			9.2	UV/VIS	[14,25]
		Quercetin-3-*O*-sophoroside	0.57	Fluorescence	[14,25]
		Quercetin-3-*O*-galactoside	<5	NMR	[25]
		Quercetin-3-*O*-glucoside	288.4	NMR	[25]
		Fisetin	14.3	UV/VIS	[25]
		Myricetin	14.6	NMR	[25]
		Naringenin	60.6	UV/VIS	[25]
	Bet v 1m	Quercetin	65.8	NMR	[25]
			26.5	UV/VIS	[25]
		Quercetin-3-*O*-galactoside	<5	NMR	[25]
		Quercetin-3-*O*-glucoside	<5	NMR	[25]
		Fisetin	68.6	UV/VIS	[25]
		Myricetin	99.3	NMR	[25]
		Naringenin	28.1	UV/VIS	[25]
	Bet v 1d	Quercetin	10.2	UV/VIS	[25]
		Fisetin	13.9	UV/VIS	[25]
		Myricetin	1.2	UV/VIS	[25]
		Naringenin	37.7	UV/VIS	[25]
Strawberry	Fra a 1E	Quercetin-3-*O*-glucuronide	5.3	ITC	[13]
	Fra a 2	Myricetin	19.5	ITC	[13]
	Fra a 3	(+)-Catechin	8.9	ITC	[13]
Hazelnut	Cor a 1	Quercetin-3-*O*-(2-*O*-β-D-glucopyranosyl)-β-D-galactopyranoside	<5	NMR	[26]
Peach	Pru p 1.0101	Zeatin	9.4	ITC	[27]

## Data Availability

Data sharing not applicable.

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
