# Peer review of "Microscale Thermophoresis Reveals Oxidized Glutathione as High-Affinity Ligand of Mal d 1"

_foods, 2021, doi:10.3390/foods10112771_

Round 1

Reviewer 1 Report

The manuscript “Microscale Thermophoresis reveals oxidized glutathione as 2 high-affinity ligand of Mal d1” seems interesting in the sense that gives more detailed information to the different ligands that may bind to a specific protein, namely Mal d 1. The manuscript is well-written, well structured, with good presentation of results and discussion. However, it has one very critical drawback, which is presented below. The authors must clearly address the critical comment and the following ones.

Critical comment: Despite the extensive presentation of Mal d 1 as an important food allergen from apple with documented cross-reactivity to Bet v 1 pollen allergen, the authors do not evaluate the effect of different ligands on the IgE-binding capacity or potential allergenicity of Mal d 1. From the point of view of structural characterisation, the manuscript can be considered of potential interest but from the clinical impact point of view, there is no correlation. Therefore, the authors need to discuss and correlate their results with potential increased or decreased allergenicity of Mal d 1 depending on the ligand bound. If possible, the Mal d 1 bound to different ligands should be tested with specific antibodies (immunoblotting or ELISA) or with apple allergic-patients’ sera. This would enable correlating the structural characterisation and the effect of ligands on the immunoreactivity/IgE-binding capacity of Mal d 1.

Other comments:

Correct to Mal d 1 along the manuscript as recommended by nomenclature of allergens. Correct the other allergen names accordingly (e.g. Bet v 1).

Lines 29-31 - provide adequate literature. Please consult Costa et al., 2020 https://doi.org/10.1007/s12016-020-08810-9 and add info.

Line 97-98 – what was the reasoning for using this extinction coefficient. Please provide adequate reference or justification.

Line 116 - correct to concentration.

Results - Did the authors try to evaluate the behaviour of natural Mal d 1 towards different ligands? Please clarify.

Lines 233-235 - Did the authors evaluate the effect of ligand binding on the secondary/3D structures of Mal d 1?

Author Response

Thank you for your comments that helped us improve this manuscript.

The manuscript “Microscale Thermophoresis reveals oxidized glutathione as high-affinity ligand of Mal d 1” seems interesting in the sense that gives more detailed information to the different ligands that may bind to a specific protein, namely Mal d 1. The manuscript is well-written, well structured, with good presentation of results and discussion. However, it has one very critical drawback, which is presented below. The authors must clearly address the critical comment and the following ones.

Critical comment: Despite the extensive presentation of Mal d 1 as an important food allergen from apple with documented cross-reactivity to Bet v 1 pollen allergen, the authors do not evaluate the effect of different ligands on the IgE-binding capacity or potential allergenicity of Mal d 1. From the point of view of structural characterisation, the manuscript can be considered of potential interest but from the clinical impact point of view, there is no correlation. Therefore, the authors need to discuss and correlate their results with potential increased or decreased allergenicity of Mal d 1 depending on the ligand bound. If possible, the Mal d 1 bound to different ligands should be tested with specific antibodies (immunoblotting or ELISA) or with apple allergic-patients’ sera. This would enable correlating the structural characterisation and the effect of ligands on the immunoreactivity/IgE-binding capacity of Mal d 1

Response: Thank you very much for pointing this out. An additional comprehensive section on this topic has been added to the discussion (lines 485-518).

[Comment] Correct Mal d 1 along the manuscript as recommended by nomenclature of allergens. Correct the other allergen names accordingly (e.g. Bet v 1).

Response: We have revised the manuscript accordingly.

[Comment] Lines 29-31 – provide adequate literature. Please consult Costa et al. 2020 and add info.

Response: We have added suitable references.

[Comment] Line 97-98 – what was the reasoning for using this extinction coefficient. Please provide adequate reference or justification.

Response: The calculation of protein concentration by UV-light absorption at 280 nm depends on the tryptophan, tyrosine and cysteine content of the protein. Therefore, an extinction coefficient was calculated based on the weighted sum of the absorption coefficients of these three amino acid residues (Trp, Tyr, Cys) according to the amino acid sequence of rMal d 1 and was then divided by the molecular weight of the protein. Adequate references were added to line 99.

[Comment] Line 116- correct to concentration.

Response: The typo has been revised accordingly.

[Comment] Results- Did the authors try to evaluate the behaviour of natural Mal d 1 towards different ligands? Please clarify.

Response: No experiments have been carried out so far to evaluate the binding of natural Mal d 1. We have added and additional sentence to the result section, that only the binding behaviour of recombinant rMal d 1 was examined (lines 181-182).

[Comment] Lines 233-235 – Did the authors evaluate the effect of ligand binding on the secondary/3D structures of Mal d1?

Response: We have not yet studied whether the ligand binding results in conformational changes in the secondary structure. Further experiments are needed to clarify if secondary structure changes are triggered by ligand binding.

Reviewer 2 Report

Chebib & Schwab describe in the present manuscript the results of an investigation of flavonoids and glutathione (both GSH and GSSG) as possible ligands for Mal d1, the major allergen found in apples. The data obtained with microscale thermophoresis and molecular docking studies allow to conclude that both GSH forms are good ligands, which provides insights into the possible mechanisms behind allergenicity of Mal d1 and other PR-10 proteins. The study has been adequately planned and conducted, presentation of data in figures is appropriate, the writing style is accurate and clear. A certain dysequilibrium is appreciable in the text as far as the discussion od data, which is at present occupying a notable space in the Results section. In order to improve readability of the paper, all considerations and comments could advantageously grouped in the Discussion section.

Author Response

Thank you for your helpful comments.

Chebib & Schwab describe in the present manuscript the results of an investigation of flavonoids and glutathione (both GSH and GSSG) as possible ligands for Mal d1, the major allergen found in apples. The data obtained with microscale thermophoresis and molecular docking studies allow to conclude that both GSH forms are good ligands, which provides insights into the possible mechanisms behind allergenicity of Mal d1 and other PR-10 proteins. The study has been adequately planned and conducted, presentation of data in figures is appropriate; the writing style is accurate and clear. A certain dysequilibrium is appreciable in the text as far as the discussion of data, which is at present occupying a notable space in the Results section. In order to improve readability of the paper, all considerations and comments could advantageously grouped in the Discussion section.

Response: Thank you very much for the suggestion. Comments and considerations have largely been removed from the results section to the discussion section.

Round 2

Reviewer 1 Report

The authors have addressed most of reviewer's comments.